# Electrospun SnO_2_/WO_3_ Heterostructure Nanocomposite Fiber for Enhanced Acetone Vapor Detection

**DOI:** 10.3390/polym15214318

**Published:** 2023-11-03

**Authors:** Ting-Han Lin, Yin-Hsuan Chang, Ting-Hung Hsieh, Yu-Ching Huang, Ming-Chung Wu

**Affiliations:** 1Department of Chemical and Materials Engineering, Chang Gung University, Taoyuan 33302, Taiwan; cgu.tinghanlin@gmail.com (T.-H.L.); cgu.yinhsuanchang@gmail.com (Y.-H.C.); a49046@gmail.com (T.-H.H.); 2Department of Materials Engineering, Ming-Chi University of Technology, New Taipei City 24301, Taiwan; 3Center for Sustainability and Energy Technologies, Chang Gung University, Taoyuan 33302, Taiwan; 4Division of Neonatology, Department of Pediatrics, Chang Gung Memorial Hospital at Linkou, Taoyuan 33305, Taiwan

**Keywords:** volatile organic compounds, chemical sensors, nanoparticles, electrospinning, sensing fibers

## Abstract

Volatile organic compounds (VOCs), often invisible but potentially harmful, are prevalent in industrial and laboratory settings, posing health risks. Detecting VOCs in real-time with high sensitivity and low detection limits is crucial for human health and safety. The optical sensor, utilizing the gasochromic properties of sensing materials, offers a promising way of achieving rapid responses in ambient environments. In this study, we investigated the heterostructure of SnO_2_/WO_3_ nanoparticles and employed it as the primary detection component. Using the electrospinning technique, we fabricated a sensing fiber containing Ag NPs, poly(methyl methacrylate) (PMMA), and SnO_2_/WO_3_ (PMMA-Ag-SnO_2_/WO_3_) for acetone vapor detection. Following activation via UV/ozone treatment, we observed charge migration between WO_3_ and SnO_2_, resulting in a substantial generation of superoxide radicals on SnO_2_ nanoparticles. This phenomenon facilitates structural deformation of the fiber and alters the oxidation state of tungsten ions, ultimately leading to a significant change in extinction when exposed to acetone vapor. As a result, PMMA-Ag-SnO_2_/WO_3_ fiber achieves a detection limit of 100 ppm and a response time of 1.0 min for acetone detection. These findings represent an advancement in the development of sensitive and selective VOC sensing devices.

## 1. Introduction

Volatile organic compounds (VOCs), predominantly colorless, can cause severe harm to the human body. VOCs are widely detected in industrial production and laboratory environments. The danger they pose is noteworthy, as their invisibility can lead to unintentional overexposure. Humans subjected to excessive VOCs can experience symptoms such as fatigue, headaches, and nervous system damage, thereby casting a shadow on overall health and well-being [1,2,3]. Acetone, a highly irritating compound, could pose significant health risks with excessive inhalation. Prolonged or high-dose exposure can irritate the eyes, nose, throat, and lungs, and cause symptoms such as dizziness, confusion, increased heart rate, unconsciousness, and even coma due to its depressant effect on the central nervous system. In addition, acetone can be found in the exhaled breath of those who suffer from diabetes. In light of this, the development of precise and responsive acetone sensors is of great significance to human health and industrial safety.

A large fraction of gas-sensing techniques is based on resistive gas sensors [4,5,6,7,8], the infrared absorption method [9,10,11,12], optical sensors [13,14,15], acoustic sensors [16,17,18,19], etc. Resistive gas sensors are highly praised for their high sensitivity, which can effectively convert changes in the gas sensing environment into unique resistance signals, thus enabling accurate detection of various gases. So far, many of them do operate effectively at room temperature, which goes against the common belief that resistive gas sensors generally require high operating temperatures. However, despite these advances, the manufacturing and operating costs associated with resistive gas sensors remain relatively high compared to other sensing technologies, primarily due to the complexity of their material synthesis, the need for specialized equipment, or, in more demanding cases, the requirement of high temperatures, increasing energy consumption [20,21]. To date, optical gas sensors have become widespread in recent years due to their unique properties. Optical VOC sensors can operate at room temperature, are highly resistant to electromagnetic radiation, and are portable and simple to use compared to traditional electrical resistive gas sensors, which require a heating component for high-temperature functioning [22,23,24,25]. Given the advances in nanomaterial technology combined with innovative synthesis methods, it has been possible to create affordable and small sensing devices with a higher sensitivity and selectivity for acetone. Nanoscale metal oxides have been widely applied as susceptible sensing materials owing to their outstanding optical and electrical properties. SnO_2_ has been widely applied in VOC sensors, which can respond to acetone, ethanol, formaldehyde, etc. [26,27,28]. Porous nanospheres of SnO_2_, catalyzed by Co_3_O_4_, have been showcased as an effective material for acetone sensing. The presence of Co_3_O_4_ introduces additional active sites that facilitate sensing reactions. Furthermore, the p-n heterointerface created between Co_3_O_4_ and SnO_2_ promotes rapid sensing reactions on the surface of the material [29]. WS_2_/WO_3_ p-n heterojunctions were constructed, exhibiting a high sensitivity to acetone owing to their increased carrier density and rapid charge transfer, able to detect acetone in a wide range from 20 to 500 ppm [30].

Among various techniques for nanomaterial synthesis, electrospinning offers significant advantages in the fabrication of gas sensors. The primary benefit is the high surface-to-volume ratio of electrospun nanofibers, which increases the exposure of reactive sites, thereby enhancing the sensitivity and response time in gas detection. This structural property facilitates a more efficient and effective adsorption process for gas molecules, crucial for achieving lower detection limits. Additionally, the porosity and interconnectedness of electrospun networks promote diffusion pathways for gas molecules, ensuring rapid sensor responses and recovery times. Moreover, electrospinning allows for the incorporation of various functional materials into the nanofibers, such as polymers, composites, and nanoparticles, which can be precisely engineered to detect specific gases with high selectivity [31]. Our group has developed a series of electrospun-fiber-based materials for VOC sensing [32,33,34,35,36,37]. The mechanism of our sensing material is based on detecting the change in extinction that contributes to the collapse of the 3D fiber structure after VOC exposure. The P3HT/PMMA fibrous film can achieve the detection limits of acetone, toluene, and o-xylene at 500 ppm [35]. Ag/CdSe-CdS/PMMA freestanding film with double-slit UV/ozone etching enabled an advanced VOC-sensing efficiency and reached the detection limit for 100 ppm of butanol within 1 min [36]. WO_3_/SnO_2_/Ag/PMMA can detect 20 ppm of acetone vapor within 10 min under room temperature, which shows potential for exhaled breath detection. The interfaces between SnO_2_ and WO_3_ might construct n–n junctions, which could help to improve the sensitivity of sensing materials when they are subjected to acetone surroundings [37]. In this regard, we would like to more thoroughly investigate the mechanism underlying acetone detection.

In this study, we synthesized a SnO_2_-decorated WO_3_ heterostructure nanocomposite using the solvothermal method to design a heterostructure for constructing n–n junctions. The in-depth characteristics of the heterostructure of SnO_2_/WO_3_ NPs were investigated. Moreover, the PMMA-Ag-SnO_2_/WO_3_ was then electrospun with Ag NPs and poly (methyl methacrylate) (PMMA) to form a sensing fiber film for acetone detection. Contributing to the gasochromic property of WO_3_, the optical properties showed an obvious difference after exposure to acetone. Based on contact potential difference measurement, we revealed a possible charge migration between WO_3_ and SnO_2_ under UV LED illumination. Based on the correlation between the optical properties and photo-assisted contact potential differences when exposed to acetone, a detection mechanism is proposed. The PMMA-Ag-SnO_2_/WO_3_ fibers successfully achieved a detection limit of 100 ppm for acetone.

## 2. Materials and Methods

### 2.1. Preparation of Sensing Materials

Silver nanoparticles (Ag NPs) were prepared through a solution-based chemical reduction reaction [38]. Amounts of 0.6 mmol of silver nitrate (AgNO_3_, >99.5%, Sigma-Aldrich, St. Louis, MO, USA) and 0.6 mmol of oleylamine (C_18_H_37_N, >98%, Sigma-Aldrich, St. Louis, MO, USA) were dissolved in 50.0 mL of chlorobenzene (C_6_H_5_Cl, >99%, ACROS, Geel, Belgium) in a four-necked flask. The mixture underwent a heating process at 120 °C for 1 h with vigorous stirring under a nitrogen flow. After cooling to room temperature, the Ag NP colloid solution was obtained. For the WO_3_@SnO_2_ NP preparation, commercial tungsten oxide (WO_3_, 99.9%, US Research Nanomaterials Inc., Houston, TX, USA) powder with an average size of 60 nm was used as the template and was decorated with SnO_2_ using the solvothermal method. Amounts of 1.0 g of SnCl_4_·5H_2_O (>98.0%, Nacalai Tesque, Kyoto, Japan) and 1.0 g of WO_3_ NPs were dispersed in 80.0 mL of benzyl alcohol (C_7_H_8_O, 99%, ACROS, Geel, Belgium) with stirring for 10 min. Subsequently, the mixture solution was transferred into a 100.0 mL Teflon-lined autoclave and heated at 120 °C for 12 h. The resulting powders were subjected to three wash cycles using diethyl ether and then collected through centrifugation at 4500 rpm for 5.0 min.

To fabricate electrospun PMMA-Ag-SnO_2_/WO_3_ fibers, a homogeneous precursor solution containing 1.0 wt.% Ag and 5.0 wt.% poly (methyl methacrylate) (PMMA, MW∼120,000 Da, Sigma-Aldrich, St. Louis, MO, USA) in dimethylformamide (DMF, C_3_H_7_NO, >99.5%, Fisher Scientific, Fair Lawn, NJ, USA) was prepared at first. Then, 20.0 mg of SnO_2_/WO_3_ NPs was added to the precursor solution. The PMMA-Ag-SnO_2_/WO_3_ precursor solution was subjected to continuous stirring at 80 °C for 3 h, ensuring complete dissolution of the PMMA. Next, the PMMA-Ag-SnO_2_/WO_3_ fibers were fabricated via the electrospinning technique. The electrospinning apparatus comprised a syringe pump (KDS−100, KD Scientific Inc., Holliston, MA, USA), a high-voltage power supply (SC−PME50, Cosmi Global Co., Ltd., New Taipei City, Taiwan), and a grounded rotary collector with dimensions measuring 15.0 cm in diameter and 15.0 cm in width (FES−COS, Falco Tech Enterprise Co., Ltd., New Taipei City, Taiwan). An applied voltage of 10.0 kV, a working distance of 10.0 cm, a flow rate of 0.5 mL/h, a solution volume of 10 mL, and a rotation rate of the collector plate of 500 rpm were employed. To activate the surface of sensing fibers, PMMA-Ag-SnO_2_/WO_3_ fiber was subjected to a 10 min dry etching through UV/ozone treatment (UV/Ozone, IAST0001−020, STAREK Scientific Co., Taipei City, Taiwan). The UV/ozone apparatus was equipped with four UVC lamps (PL−L, 36W, λ max = 254 nm, Philips, Amsterdam, The Netherlands).

### 2.2. Material Characterization

The crystal structure analysis of WO_3_ and SnO_2_/WO_3_ NPs was conducted using X-ray diffractometer to record the diffraction patterns of as-prepared nanomaterials. Spherical-aberration-corrected field emission transmission electron microscope (ULTRA-HRTEM, JEM-ARM200FTH, JEOL, Tokyo, Japan) equipped with Si drift detector for elemental analysis was employed to observe the morphology and elemental distribution. The size distribution of particles was recorded and calculated from individual measurements of at least 100 particles. The contact potential difference was measured using a Kelvin probe analyzer (SKP 5050, KP technology, Wick, Caithnes, UK). The morphology of the fibers was observed using field-emission scanning electron microscope (FESEM, model SU8010, Hitachi, Tokyo, Japan). UV−VIS spectrophotometer (UV−1900i, Shimadzu, Kyoto, Japan) was used to examine the optical properties of the fibers.

### 2.3. Detection of Acetone Vapor

To examine the sensitivity of PMMA-Ag-SnO_2_/WO_3_ sensing fiber to acetone vapor, the sensing chip was placed in the middle of the container for the acquisition of extinction measurement. The specific volume of acetone solution was injected into a 100.0 mL glass liner. The volume for various concentrations of acetone vapor production was calculated using the ideal gas equation of state (PV = nRT). Following evaporation, we sampled the concentrated acetone vapor and transferred it into a 4.5 cm by 4.0 cm by 4.0 cm quartz glass container. At the same time, we calculated the volume needed to dilute the concentrated vapor. Subsequently, upon injecting the acetone vapor, we initiated measurements of the extinction spectra. The extinction spectrum was immediately recorded using a UV-VIS spectrometer, scanning wavelengths from 400 nm to 900 nm, with measurements taken every 30 s. The examination of acetone detection was conducted at room temperature (25 °C) within a relative humidity range of 60%. In general, the transmittance of the materials results in various extinction intensities. PMMA-Ag-SnO_2_/WO_3_ fiber exhibits a broad extinction shoulder at around 410 nm. Hence, we defined the change in extinction as the difference in extinction peak observed before and after exposure to VOC vapor. The equation of extinction change (∆E_t_) is described as follows:∆E_t_ = E_t_ − E_0_(1)
where E_0_ represents the extinction before exposure to VOCs and E_t_ represents the extinction intensity after exposure to VOC vapor at a specific time point, t mins.

## 3. Results

The heterostructure of WO_3_ and SnO_2_ was confirmed using X-ray diffraction analysis and via observation using an electron microscope. Figure 1a shows the X-ray diffraction patterns of pristine WO_3_ NPs and SnO_2_/WO_3_ NPs. The typical diffraction of WO_3_ is observed in both nanoparticles, which is located at 2θ of 23.15°, 23.64°, 24.33°, 26.56°, 28.93°, 33.56°, 34.11°, 41.91°, 49.87°, 55.30°, and 61.75° indexed to the lattice planes of (001), (020), (200), (120), (111), (021), (220), (221), (104), (402), and (340) [JCPDS 00-005-0363], respectively. A weak diffraction of SnO_2_ in SnO_2_/WO_3_ NPs can be indexed to (200) and (211) at 2θ of 38.96° and 51.90° [JCPDS 00-021-1250], respectively. With the incorporation of SnO_2_ and undergoing the solvothermal reaction, the diffraction intensity of WO_3_ decreased dramatically, attributed to the disintegration of the aggregated WO_3_ NPs and slight doping by Sn [39]. Figure 1b,c show the morphology of WO_3_ NPs and SnO_2_/WO_3_ NPs, and ultra-small particles are especially decorated on WO_3_ NPs. To confirm the presence and composition of SnO_2_, we further investigated their microstructure and detailed surface information via a transmission electron microscope. The aggregated WO_3_ showed a bare surface (Figure 1d) with a diameter of around a hundred nanometers, and plenty of nanoparticles were covered when incorporated with SnO_2_ (Figure 1e). In the magnified image of Figure 1f, we were able to index the lattice fringe for WO_3_ with a d-spacing of 3.61 Å for (220) in the core template and SnO_2_ with a d-spacing of 3.36 Å for (110) on the surface as a shell. It depicts the successful incorporation of SnO_2_ onto WO_3_ NPs and establishes the heterostructure. The elemental mapping and estimated composition of SnO_2_/WO_3_ are shown in Figure 1g–l. The good distribution of Sn covering the WO_3_ NPs is realized and the composition is 59.0% ± 0.8, 34.4% ± 0.7, and 6.7% ± 0.5 for tungsten, oxygen, and tin, respectively. In brief summary, the nanoscale heterostructure of nano-sized SnO_2_ decorated on WO_3_ NPs can be achieved via hydrothermal reaction.

We then incorporated the as-prepared SnO_2_/WO_3_ NPs into the PMMA and Ag NP precursor solution and proceeded to fabricate a sensing fiber using the electrospinning technique. A photograph of the PMMA-Ag-SnO_2_/WO_3_ sensing chip is illustrated in Appendix A. Pure PMMA-Ag fibers present a regular and uniform fiber morphology with an average diameter of 198 ± 37 nm, and a smooth surface (Figure 2a,b). However, with the incorporation of SnO_2_/WO_3_ NPs, the interconnected network structure is reduced, and some beads become embedded into the PMMA-Ag fibers. It leads to the average diameter, and the relevant deviation in the calculated diameter increases to 204 ± 88 nm (Figure 2c,d). Through EDS mapping observation, we confirmed the presence of embedded particles composed of WO_3_ and SnO_2_, as well as the random distribution of SnO_2_/WO_3_ NPs in the fibers (Figure 2e–i). In addition, UV/ozone treatment was employed to activate the surface of composite fibers before the acetone detection. The outward appearance of the sensing chip after UV/ozone treatment shows an insignificant change (Appendix A). To confirm the preservation of the three-dimensional network structure of composite fibers after 10 min of UV/ozone treatment, observation of the morphology was carried out, which is shown in Appendix A. The fibers demonstrated slight shrinkage and their surfaces became rough. Importantly, the integrity of the three-dimensional network structure remained unaltered. This suggests that ozone molecules and oxygen radicals engaged in a mild reaction with the surface of composite fibers.

To evaluate the VOC detection of PMMA-Ag-SnO_2_/WO_3_ sensing fibers, we measured the transmittance using a UV-Vis spectrometer and converted it into extinction spectra. The change in maximum extinction over time corresponds to the detection response. Figure 3a presents the extinction spectra of PMMA-Ag-SnO_2_/WO_3_ fibers exposed to 10,000 ppm of acetone. A significant downshift in extinction appears at wavelengths ranging from 400 to 500 nm, inferring the fibers’ structure deformation and WO_3_ gasochromic properties when exposed to acetone vapor. The Ag-induced SPR effect magnified the trend of decreased extinction. The outward appearance also exhibits a slight fade when the sensing chip is exposed to acetone (Appendix A). As for the control sample of PMMA-Ag-WO_3_ fibers (Figure 3b), the maximum ∆E when exposed to air over time is approximately −0.008. This value serves as the threshold for acetone detection to mitigate the influence of environmental factors. The response time is identified when the change in extinction (∆E) at a given exposure time (t) exceeds the threshold. Furthermore, both PMMA-Ag-WO_3_ fibers and PMMA-Ag-SnO_2_/WO_3_ fibers (Figure 3b,c) exhibit a noticeable ∆E when exposed to acetone vapor. The change in ∆E over time for PMMA-Ag-SnO_2_/WO_3_ fibers, when exposed to acetone concentrations of 10,000, 1000, and 100 ppm, is higher compared to that of PMMA-Ag-WO_3_ fibers. This indicates that the incorporation of SnO_2_ NPs to fabricate the heterostructured SnO_2_/WO_3_ NPs enhances the extinction response and sensitivity of acetone vapor. The reproducibility of the sensing fiber was further validated. The sensing fibers displayed highly consistent changes in extinction, as shown in Appendix A. To further realize the lowest concentration of detection, we collected the several changes in extinction of sensing fibers exposed to acetone vapor with various concentrations and depicted the calibration curve (Appendix A). A positive correlation was observed between the vapor concentration and the change in extinction (Appendix A), indicating a strong dependence of the change in extinction on the concentration of acetone vapor. Additionally, through the formulation of vapor concentration and the corresponding change in average extinction at t = 30 min, the calibration curve can be defined as follows:ΔE_t_ = −0.00516 × ln(C) − 0.016(2)
where ΔE represents the change in extinction at a given exposure time (t), and C denotes the vapor concentration in ppm. Considering the threshold (−0.008) of the change in extinction in ambient air, the sensing fiber is anticipated to exhibit sensitivity down to a concentration of 0.212 ppm. Consequently, the expected detection limit and response time for acetone detection using PMMA-Ag-SnO_2_/WO_3_ fibers are notably below 100 ppm and within 1.0 min, respectively.

To investigate the enhancement in VOC detection achieved through the incorporation of SnO_2_/WO_3_ NPs and its rapid response to acetone vapor, we speculated that the n-n heterojunction of WO_3_ and SnO_2_ are crucial factors, resulting in the strong change in extinction. Its interfacial charge migration after contact was investigated using a Kelvin probe force analyzer to qualify the change in contact potential difference with UV irradiation. To enhance the response of VOC detection, it is crucial to undergo a UV/ozone pretreatment, which activates both the polymer fibers and SnO_2_/WO_3_ NPs. Therefore, we employed a UV-LED as a light source to activate the semiconductor when measuring the CPD. Figure 4 shows the equilibrium contact potential difference (CPD) of WO_3_ and the SnO_2_/WO_3_ heterostructure nanocomposite with and without UV LED irradiation. The average CPDs of WO_3_ NPs without and with illumination are 280.39 ± 6.29 mV and 136.75 ± 3.81 mV, respectively (Figure 4a,b). The bare WO_3_ NPs exhibit a significant change in CPD, implying the presence of rich electrons on the surface of WO_3_ NPs under UV LED irradiation. Through light on/off cycle tests, the dynamic changes in CPD also indicate the photoresponse property of WO_3_ NPs, with an average difference of approximately 15.57 mV. Additionally, the growth of SnO_2_ NPs onto WO_3_ NPs reduces the equilibrium CPD difference, resulting in 315.38 ± 2.99 mV without illumination and 224.35 ± 2.42 mV with illumination (Figure 4d,e). The dynamic changes in CPD show a similar trend, decreasing to 13.26 mV (Figure 4f). We deduced that under UV LED illumination, the photo-induced electrons generated by WO_3_ can transfer to SnO_2_ and may subsequently react with O_2_ to promptly generate superoxide radicals, leading to their depletion. The significant depletion of electrons can be attributed to SnO_2_ being an intrinsic n-type semiconductor with a high density of oxygen vacancies, which leads to a preference for oxygen adsorption [39].

Based on the observation of changes in CPD with and without illumination, we believe that the aforementioned phenomena also occur when the SnO_2_/WO_3_ nanoparticles undergo UV/ozone treatment. The possible mechanism is provided as follows. When SnO_2_/WO_3_ NPs are exposed to the UV/ozone, photo-induced electrons pass through the interface and transfer from WO_3_ to SnO_2_, further reacting with O_2_ to promptly generate superoxide radicals (O_2_^−^) (Equation (3)). These radicals either adsorb to PMMA or reduce the hexavalent tungsten to pentavalent tungsten (Equations (4) and (5)). The radicals activate the surface of PMMA but weaken the fibers’ structure. When activated PMMA-Ag-SnO_2_/WO_3_ fibers are exposed to acetone vapor, the swelling effect diminishes the light scattering of the weakened fiber structure, leading to significant structural deformation and a subsequent decrease in extinction. Furthermore, the incorporation of Ag NPs with surface plasmon resonance further magnifies the decreased extinction behavior, thereby enhancing sensitivity (Figure 3a). On the other hand, under the acetone atmosphere, the collapsed PMMA fibers cannot provide sufficient surface-adsorbed O_2_^−^ radicals, and radicals are also exhausted due to acetone degradation (Equations (6)–(8)). These behaviors disrupt the reduction of hexavalent tungsten and cause pentavalent tungsten to release electrons to SnO_2_ due to the equilibrium heterojunction, converting back to hexavalent tungsten. Consequently, the extinction at wavelengths around 500–600 nm, attributed to WO_3_, increases, offsetting the overall downward trend. To confirm the mechanism, we further used the chemical inert gas, n-hexane, for sensing the fiber response. The corresponding extinction spectra and change in extinction at varying exposure times are shown in Appendix A. The extinction spectra of sensing fibers remained unaltered upon exposure to n-hexane vapor, suggesting that n-hexane neither deforms the sensing fibers nor influences the light scattering of the sensing fiber. Moreover, it does not trigger any gasochromic behavior in heterostructured SnO_2_/WO_3_ NPs. Given that PMMA is insoluble in hexane, swelling and adsorption effects are not anticipated, thereby resulting in no observable changes in light scattering. In addition, the stable alkane structure of n-hexane is not readily influenced by radicals produced from UV/ozone treatment or photo-irradiation of SnO_2_/WO_3_. Thus, no redox reactions with SnO_2_/WO_3_ nanoparticles occur, and no gasochromic effects are observed. In summary, these observations confirm an enhanced sensitivity to acetone vapor.
(3)$$e_{{(\mathrm{WO}}_{3})}^{-}\to e_{{(\mathrm{SnO}}_{2})}^{-}+O_{2}\to {{\cdot O}_{2}^{-}}_{(\mathrm{Sn}O_{2})}$$
(4)$${{\cdot O}_{2}^{-}}_{(\mathrm{Sn}O_{2})}\to {{\cdot O}_{2}^{-}}_{\left(\mathrm{PMMA}\right)}$$
(5)$$W^{6+}+{\cdot O}_{2}^{-}\to W^{5+}+O_{2}$$
(6)$${\mathrm{CH}}_{3}{\mathrm{COCH}}_{3}+{\cdot O}_{2}^{-}\to {\mathrm{CH}}_{3}C^{+}O+{\mathrm{CH}}_{3}O^{-}+e^{-}$$
(7)$${\mathrm{CH}}_{3}C^{+}O\to {\mathrm{CH}}_{3}+\mathrm{CO}$$
(8)$$\mathrm{CO}+{\cdot O}_{2}^{-}\to {\mathrm{CO}}_{2}+e^{-}$$

## 4. Conclusions

PMMA-Ag incorporated with SnO_2_/WO_3_ heterostructure nanocomposite sensing fibers is successfully fabricated using the electrospinning technique. The characteristics of the synthesized SnO_2_/WO_3_ confirm the successful growth of SnO_2_ on WO_3_ nanoparticles via a solvothermal reaction. The morphology and elemental distribution of embedded SnO_2_/WO_3_ NPs in PMMA-Ag fibers were demonstrated. After the UV/ozone treatment, the charge migration within the WO_3_ and SnO_2_ heterojunction, as indicated by changes in the contact potential difference under UV illumination, leads to the substantial generation of superoxide radicals. This phenomenon enhances the change in extinction and detection response over time. As a result, the PMMA-Ag-SnO_2_/WO_3_ fibers achieve an acetone detection limit of 100 ppm with a response time of just 1.0 min. This research highlights noteworthy progress in the field of VOC sensing technology.

## Figures and Tables

**Figure 1 polymers-15-04318-f001:**
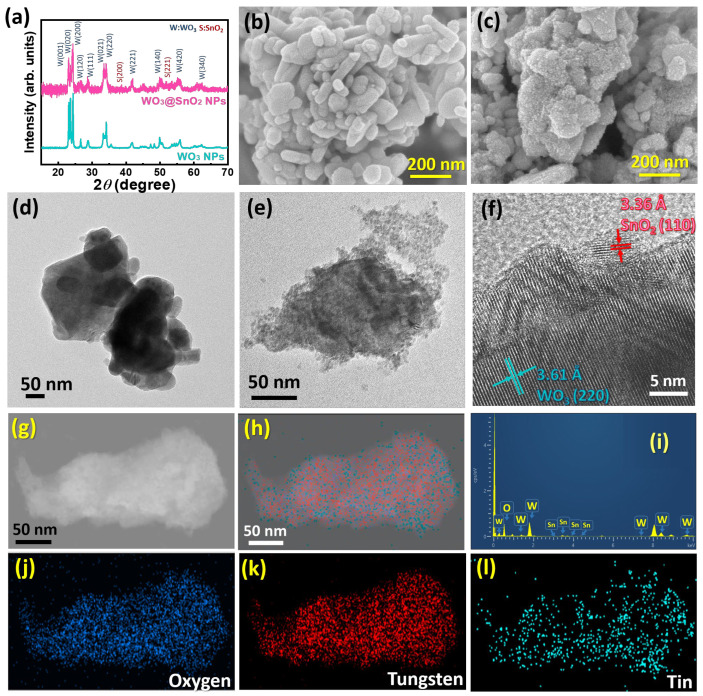
Characteristic and surface features of pristine WO_3_ NPs and as-prepared SnO_2_/WO_3_ NPs. (**a**) XRD patterns, (**b**,**c**) SEM images, and (**d**,**e**) TEM images for WO_3_ NPs and SnO_2_/WO_3_ NPs, (**f**) high-resolution TEM image of SnO_2_/WO_3_ NPs, and the related elemental dispersive spectra in mapping mode, (**g**) selected-area image of nanoparticle, (**h**) merge image, (**i**) EDS spectra, (**j**) oxygen, (**k**) tungsten, and (**l**) tin.

**Figure 2 polymers-15-04318-f002:**
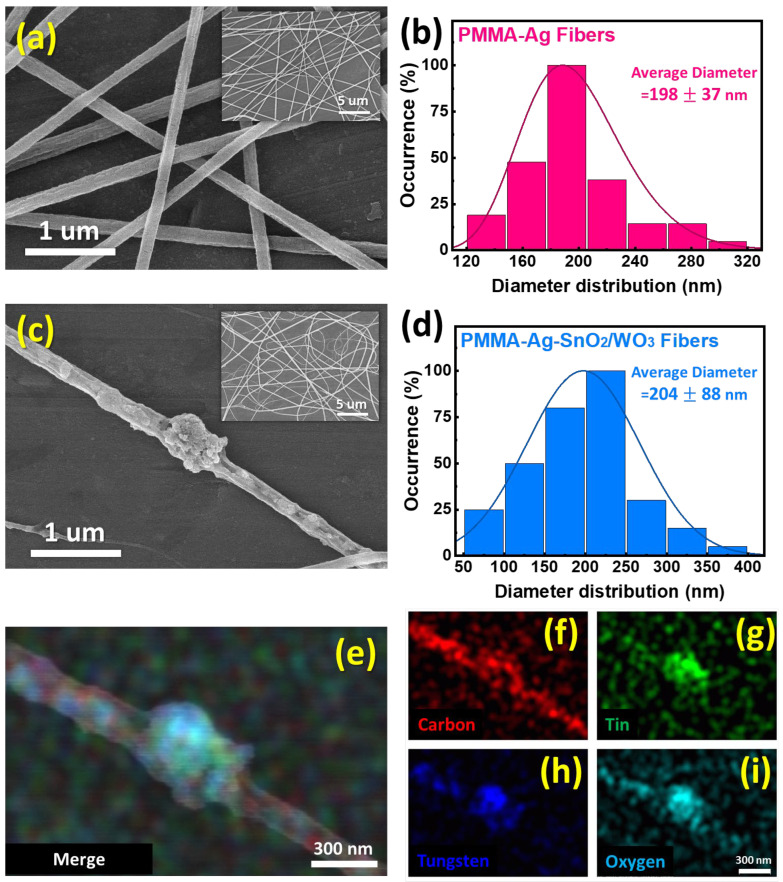
Characteristics of as-prepared sensing fibers. SEM images of (**a**) PMMA-Ag fibers and (**c**) PMMA-Ag-SnO_2_/WO_3_ fibers, inset shows related low-magnification image. The related diameter distribution of (**b**) PMMA-Ag fibers and (**d**) PMMA-Ag-SnO_2_/WO_3_ fibers. EDS mapping image of PMMA-Ag-SnO_2_/WO_3_ fibers, (**e**) merge image, (**f**) carbon, (**g**) tin, (**h**) tungsten, and (**i**) oxygen.

**Figure 3 polymers-15-04318-f003:**
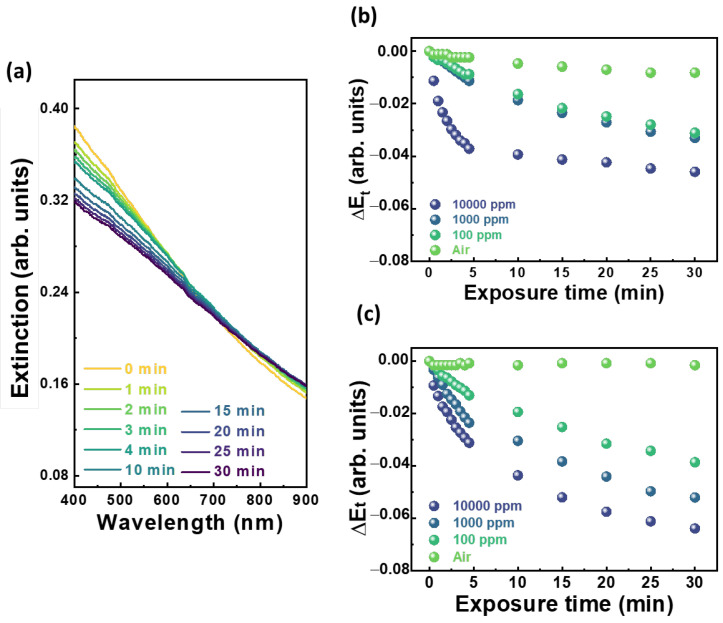
VOC detection examination. (**a**) Extinction spectra of PMMA-Ag-SnO_2_/WO_3_ fibers exposed to 10,000 ppm acetone. Extinction changes of (**b**) PMMA-Ag-WO_3_ fibers and (**c**) PMMA-Ag-SnO_2_/WO_3_ fibers exposed to acetone with various concentrations over exposure time, t min.

**Figure 4 polymers-15-04318-f004:**
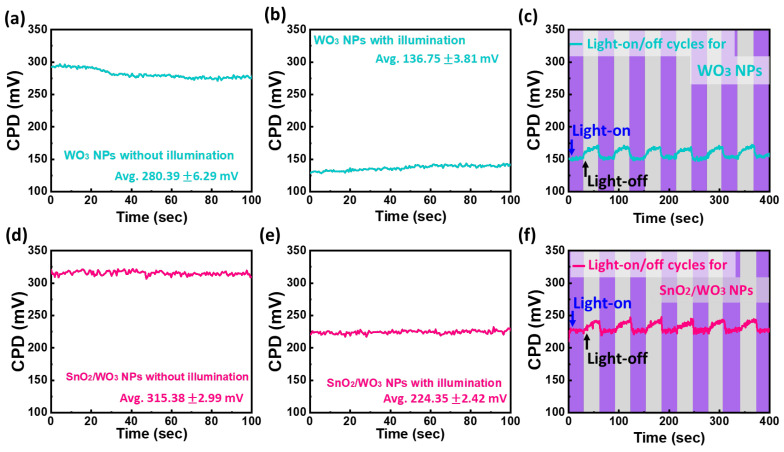
Contact potential difference (CPD) measurement via Kelvin probe force analyzer. CPD of WO_3_ NPs (**a**) without illumination, (**b**) with illumination, and (**c**) light-on/off cycles test. And CPD of SnO_2_/WO_3_ NPs (**d**) without illumination, (**e**) with illumination, and (**f**) light-on/off cycles test.

## Data Availability

Data are available from the corresponding author upon reasonable request.

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
