# Peer review of "Electrospun SnO2/WO3 Heterostructure Nanocomposite Fiber for Enhanced Acetone Vapor Detection"

_polymers, 2023, doi:10.3390/polym15214318_

Round 1

Reviewer 1 Report

Comments and Suggestions for Authors

In this paper, the authors fabricated SnO2@WO3@Ag composite materials for acetone detection. In general, the sensing properties against acetone is acceptable, but major revision is required for further consideration.

1. The novelty within the introduction part is quite poor, the authors should rewritte it to emphasize the novelty of this work.

2. The authors should provided enough charaterizations of the PMMA-Ag-SnO2/WO3 nanocomposite materials before and after 10 min dry etching by UV ozone treatment.

3. The reason on using electrospinning istead of directly using Ag coating on SnO2/WO3 nanocomposites should be also provided.

Comments on the Quality of English Language

The English of this paper should be polished to enhance the readability of the work.

Author Response

Reviewer 1
In this paper, the authors fabricated SnO2@WO3@Ag composite materials for acetone detection. In general, the sensing properties against acetone is acceptable, but major revision is required for further consideration.

  1. The novelty within the introduction part is quite poor, the authors should rewrite it to emphasize the novelty of this work.

Reply:

Thank you for your comment. We have rewrite the introduction part.

  1. The authors should provided enough charaterizations of the PMMA-Ag-SnO2/WO3 nanocomposite materials before and after 10 min dry etching by UV ozone treatment.

Reply:

Thank you for your comment. We conducted scanning electron microscopy to examine the morphology of PMMA-Ag-SnO2@WO3 fibers after 10 minutes of UV-ozone treatment. The composite fibers exhibited slight shrinkage and their surfaces became relatively rough. This suggests that ozone molecules and oxygen radicals engaged in a mild reaction with the surface of composite fibers. The image was added into the supporting information.

Figure S2. Morphology of PMMA-Ag- SnO2/WO3 fibers with UV ozone treatment.

Line 206, page 5 of the revised manuscript

We then incorporated the as-prepared SnO2/WO3 NPs into the PMMA and Ag NPs precursor solution and proceeded to fabricate a sensing fiber using the electrospinning technique. The photograph of the PMMA-Ag-SnO2/WO3 sensing chip is illustrated in Figure S1a. Pure PMMA-Ag fibers present a regular and uniform fiber morphology with an average diameter of 198 37 nm, and a smooth surface (Figure 2a-b). However, with the incorporation of SnO2/WO3 NPs, the interconnected network structure is reduced, and some beads become embedded into the PMMA-Ag fibers. It leads to the average diameter and relevant deviation of the calculated diameter increase to 204 88 nm (Figure 2c-d). Through EDS mapping observation, we confirmed the presence of embedded particles composed of WO3 and SnO2, as well as the random distribution of SnO2/WO3 NPs in the fibers. (Figure 2e-i). In addition, the UV-ozone treatment was employed to activate the surface of composite fibers before the acetone detection. The outward appearance of the sensing chip after UV-ozone treatment shows insignificant change (Figure S1b).  To confirm the preservation of the three-dimensional network structure of composite fibers after 10 mins of UV-ozone treatment, the morphology observation was undertaken, which is shown in Figure S2. The fibers demonstrated slight shrinkage and their surfaces became rough. Importantly, the integrity of the three-dimensional network structure remained unaltered. This suggests that ozone molecules and oxygen radicals engaged in a mild reaction with the surface of composite fibers.

  1. The reason on using electrospinning instead of directly using Ag coating on SnO2/WO3 nanocomposites should be also provided.

Reply:

First, I would like to thank you for your effort to help us improve the quality of our manuscript. Our material design is multifaceted and hinges on both performance optimization and structural considerations. Primarily, our sensor's functionality is intimately tied to the structural alterations in the electrospun fiber upon VOC exposure. Structural deformations trigger the detectable extinction changes, fundamental to our sensor's operation. The integration of Ag NPs is not arbitrary but a strategic enhancement. When these nanoparticles are embedded within the electrospun fibers, they exploit the SPR effect, known for its sensitivity to environmental changes. This phenomenon significantly amplifies the sensor's signal, providing a more pronounced response upon VOC interaction. Besides, incorporating different materials or dopants into the electrospun nanofibers, it is possible to enhance the selectivity of the sensor to specific VOCs of interest. This selectivity can be crucial in applications where the discrimination between different volatile organic compounds is required. Moreover, it's worth noting that direct Ag coating may lead to the shadowing effect, potentially resulting in a less efficient gas-sensing surface when exposed to SnO2/WO3 nanocomposites.

Reviewer 2 Report

Comments and Suggestions for Authors

The manuscript by Lin etal reports on the preparation, characterization and possible sensing application of a novel nanocomposite material. The material was prepared by solvothermal synthesis of SnO2 NPs on commercial WO3 nanopowder as template, followed by mixing with colloidal Ag-NPs and PMMA solution. This solution was then transformed into polymer-nanocomposite fibers by electrospinning.

These nanofibers were proposed as potential sensor material for optical VOC detection and first measurement results for exposure to acetone vapour were presented.

The parts on materials synthesis and general materials characterization by XRD and microscopic techniques are carried out very well and conclusive results are presented. These proof the formation of heterostructures of small SnO2 NP decorating the larger WO3 particles and also incorporation of the composite particles as well as the Ag-NP into the polymer nanofibers.

I am however much more skeptical about the actual sensing/analytical function and value of this material for the proposed optical VOC sensing application.

The extinction spectra presented show no specific peaks but more the common light scattering behavior of a fibrous network. The observed effect of an reduction of extinction on the short wavelength end of the spectrum is most likely caused by a quite unspecific change of light scattering due to swelling and partial collapse of the polymer fiber network under acetone exposure. And of course the scattering effect is amplified if the polymer fibers are filled with more scattering particles.

From a sensing point of view it would be absolutely necessary to add further essential information:

·         describe, whether or not the effect is reversible after removal of acetone vapour and whether if can be repeated over several cycles of VOC exposure?

·         Information on the reproducibility of the quantitative extinction reduction over various samples and experiments should be added.

·         In Fig. 3 b,c there seems to be a quite strange and non-linear “dose-response” behaviour of extinction to acetone concentration. Furthermore, it seems to change over the response time of the sensor (for short times the response for 100 ppm seems to be stronger than for higher conc.) This needs to be explained and a sort of calibration curve (effect vs. analyte concentration) needs to be established as basic information for any potential sensing setup.  

·         According to Fig. 3 b,c, the responses to 100 ppm Acetone are very much higher than the blank/baseline level. If this behavior is reproducible and stable, the detection limit would be even much lower than the 100 ppm stated in the manuscript (by using common definitions and methods for determining LODs of chemical sensors: see e.g. in H.P. Look et al., SensActB: Chem, 173(2012) 157). So, if the LOD is stated in the article , an LOD should be determined from a calibration curve and blank level or another commonly applied procedure.

·         If the measured optical effect is primarily a change in light scattering of the fiber structure it could be high sensitive also to interferences by humidity changes and this should be discussed? 

·         The sensing mechanism given in p.8 , involving redox reaction with acetone etc. is still highly speculative.  It should be e.g. backed up with a control experiment using e.g. chemical inert solvent vapour. Furthemore, a short general discussions regarding specificity to e.g. acetone over other VOC´s shall be added.

Other minor points for consideration:

·         The term “Core-Shell” Nanocomposite Fiber in the title is a bit misleading as the material is not structured as classical core-shell fibers, nor are the NP´s synthesized real core-shell particles.

·         In the EDS spectrum presented in Fig. 1i its almost impossible to read or see anything due to small image size and bad resolution. So size and resolution should be increased or the spectrum might be attached in full size as supplementary material

·         Figure 2: Given the width of fiber size distributions, its not meaningful to report the fiber diameters to 0,1 nm, significant digits (also in the text on p. 5).

·         Typo in Eq. 4: there should be pentavalent W(V) on the right side of the Eq.

Comments on the Quality of English Language

Several grammar errors and flaws. general proofreading recommended

Author Response

Reviewer 2

The manuscript by Lin etal reports on the preparation, characterization and possible sensing application of a novel nanocomposite material. The material was prepared by solvothermal synthesis of SnO2 NPs on commercial WO3 nanopowder as template, followed by mixing with colloidal Ag-NPs and PMMA solution. This solution was then transformed into polymer-nanocomposite fibers by electrospinning.

These nanofibers were proposed as potential sensor material for optical VOC detection and first measurement results for exposure to acetone vapour were presented.

The parts on materials synthesis and general materials characterization by XRD and microscopic techniques are carried out very well and conclusive results are presented. These proof the formation of heterostructures of small SnO2 NP decorating the larger WO3 particles and also incorporation of the composite particles as well as the Ag-NP into the polymer nanofibers.

I am however much more skeptical about the actual sensing/analytical function and value of this material for the proposed optical VOC sensing application.

The extinction spectra presented show no specific peaks but more the common light scattering behavior of a fibrous network. The observed effect of an reduction of extinction on the short wavelength end of the spectrum is most likely caused by a quite unspecific change of light scattering due to swelling and partial collapse of the polymer fiber network under acetone exposure. And of course the scattering effect is amplified if the polymer fibers are filled with more scattering particles.

From a sensing point of view it would be absolutely necessary to add further essential information:

  1. Describe, whether or not the effect is reversible after removal of acetone vapour and whether if can be repeated over several cycles of VOC exposure? Information on the reproducibility of the quantitative extinction reduction over various samples and experiments should be added.

Reply:

Thank you for the constructive suggestion. In terms of the recyclability of our sensing materials, it's important to note that they exhibit significant and irreversible deformation when exposed to VOCs. However, the unique performance characteristics of these sensing materials, such as high sensitivity, real-time detection, and cost-effectiveness, remain quite remarkable. Consequently, these sensing materials could be considered as disposable sensing chips suitable for practical applications.         

  1. In Fig. 3 b,c there seems to be a quite strange and non-linear “dose-response” behaviour of extinction to acetone concentration. Furthermore, it seems to change over the response time of the sensor (for short times the response for 100 ppm seems to be stronger than for higher conc.) This needs to be explained and a sort of calibration curve (effect vs. analyte concentration) needs to be established as basic information for any potential sensing setup.  

Reply:

Thank you for the constructive suggestion. We have carefully tested the acetone detection several times at varying concentrations, as shown in Figure 3 and Figure S3.
In a series of reproducibility tests, acetone detection was conducted three times, yielding highly consistent changes in extinction as shown in Figure S3. Additionally, a positive correlation was observed between the vapor concentration and the extinction change (Figure S4a), indicating a strong dependence of the extinction change on the concentration of acetone vapor. A calibration curve describing the relationship between vapor concentration and sensor responsivity is also presented (Figure S4b).

Figure S3. Reproducibility test of PMMA-Ag-SnO2/WO3 fibers exposed to 100 ppm acetone vapor.

Line 232, page 6 of the revised manuscript

To evaluate VOCs detection of PMMA-Ag-SnO2/WO3 sensing fibers, we measured the transmittance using a UV-Vis spectrometer and converted it into extinction spectra. The change in maximum extinction over time corresponds to the detection response. Figure 3a presents the extinction spectra of PMMA-Ag-SnO2/WO3 fibers exposed to 10,000 ppm acetone. The significant downshift of extinction appears at wavelengths ranging from 400 to 500 nm, inferring the fibers' structure deformation and WO3 gaschromic properties as exposed to acetone vapor. The Ag-induced SPR effect magnified the trend of decreased extinction. The outward appearance also exhibits a slight fade when the sensing chip is exposed to acetone (Figure S1c). As for the control sample of PMMA-Ag-WO3 fibers (Figure 3b), the maximum ∆E when exposed to air over time is approximately -0.008. This value serves as the threshold for acetone detection to mitigate the influence of environmental factors.  The response time is identified when the extinction change (∆E) at a given exposure time (t) exceeds the threshold. Furthermore, both PMMA-Ag-WO3 fibers and PMMA-Ag-SnO2/WO3 fibers (Figure 3b,c) exhibit a noticeable ∆E as exposed to acetone vapor. The change in ∆E over time for PMMA-Ag-SnO2/WO3 fibers, when exposed to acetone concentrations of 10,000, 1,000, and 100 ppm, is higher compared to that of PMMA-Ag-WO3 fibers. It indicates the incorporation of SnO2 NPs to fabricate the heterostructured SnO2/WO3 NPs enhances the extinction response and sensitivity of acetone vapor. The reproducibility of the sensing fiber was further validated. The sensing fibers displayed highly consistent changes in extinction, as shown in Figure S3

  1. According to Fig. 3 b,c, the responses to 100 ppm Acetone are very much higher than the blank/baseline level. If this behavior is reproducible and stable, the detection limit would be even much lower than the 100 ppm stated in the manuscript (by using common definitions and methods for determining LODs of chemical sensors: see e.g. in H.P. Look et al., SensActB: Chem, 173(2012) 157). So, if the LOD is stated in the article , an LOD should be determined from a calibration curve and blank level or another commonly applied procedure.

Reply:

Thank you for the constructive suggestion. We have gathered data on the extinction curves for sensing fibers across various concentrations of acetone vapor and have consequently formulated a calibration curve, which is displayed in Figure S4. The equation describing the curve is as follows:

Et= -0.00516 ln(C) -0.016

Where ΔE represents the change in extinction at a given exposure time (t), and C denotes the vapor concentration in ppm. Considering the threshold (-0.008) of extinction change in ambient air, the as-prepared sensing fiber is anticipated to exhibit sensitivity down to a concentration of 0.212 ppm.

Figure S4. (a) Extinction change of PMMA-Ag-SnO2/WO3 fibers exposed to acetone with various concentration ppm over exposure time, t min and (b) the calibration curve corresponding to 30-min exposure of PMMA-Ag-SnO2/WO3 fibers to acetone vapor.

Line 251, page 7 of the revised manuscript

To further realize the lowest concentration of detection, we collected the several extinction changes of sensing fibers exposed to acetone vapor with various concentrations and depicted the calibration curve (Figure S4). A positive correlation was observed between the vapor concentration and the extinction change (Figure S4a), indicating a strong dependence of the extinction change on the concentration of acetone vapor. Additionally, through the formulation of vapor concentration and the corresponding average extinction change at t = 30 mins, the calibration curve can be defined as follows:

Et= -0.00516 ln(C) -0.016                                                                  (2)

Where ΔE represents the change in extinction at a given exposure time (t), and C denotes the vapor concentration in ppm. Considering the threshold (-0.008) of extinction change in ambient air, the sensing fiber is anticipated to exhibit sensitivity down to a concentration of 0.212 ppm. Consequently, the expected detection limit and response time for acetone detection using PMMA-Ag-SnO2/WO3 fibers are notably below 100 ppm and within 1.0 min.

  1. If the measured optical effect is primarily a change in light scattering of the fiber structure it could be high sensitive also to interferences by humidity changes and this should be discussed? 

Reply:

Thank you for the review’s comment. To mitigate the impact of ambient humidity on acetone vapor detection, we initially conducted a blank test with the sensing fiber exposed to ambient atmosphere, that was labeled as air in the plot of acetone detection. The observed change in extinction during this blank test was established as the threshold level. Thus, if the extinction change of the sensing fiber at any given concentration of acetone vapor surpasses this threshold, it is indicative of a response to the acetone vapor at that specific concentration.

Line 232, page 6 of the revised manuscript

To evaluate VOCs detection of PMMA-Ag-SnO2/WO3 sensing fibers, we measured the transmittance using a UV-Vis spectrometer and converted it into extinction spectra. The change in maximum extinction over time corresponds to the detection response. Figure 3a presents the extinction spectra of PMMA-Ag-SnO2/WO3 fibers exposed to 10,000 ppm acetone. The significant downshift of extinction appears at wavelengths ranging from 400 to 500 nm, inferring the fibers' structure deformation and WO3 gaschromic properties as exposed to acetone vapor. The Ag-induced SPR effect magnified the trend of decreased extinction. The outward appearance also exhibits a slight fade when the sensing chip is exposed to acetone (Figure S1c). As for the control sample of PMMA-Ag-WO3 fibers (Figure 3b), the maximum ∆E when exposed to air over time is approximately -0.008. This value serves as the threshold for acetone detection to mitigate the influence of environmental factors.  The response time is identified when the extinction change (∆E) at a given exposure time (t) exceeds the threshold.

  1. The sensing mechanism given in p.8 , involving redox reaction with acetone etc. is still highly speculative.  It should be e.g. backed up with a control experiment using e.g. chemical inert solvent vapour. Furthemore, a short general discussions regarding specificity to e.g. acetone over other VOC´s shall be added.

Reply:

Thank you for the review’s comment. We have conducted a n-hexane detection at 10,000 ppm using the sensing fiber to observe its response. In Figure S5, the extinction spectra of sensing fibers remained unaltered upon exposure to n-hexane vapor. This behavior suggests that n-hexane does not cause the deformation and influence the light scattering of sensing fiber, nor does it the trigger the gaschromic behavior of the heterostructured SnO2/WO3 NPs. PMMA is insoluble in hexane; therefore, swelling and adsorption effects are not expected, leading to no observable changes in light scattering. In addition, the alkane structure of n-hexane is highly stable under ambient condition. Consequently, radicals generated from UV-ozone treatment or photo-irradiation of SnO2/WO3 are unlikely to significantly impact n-hexane or initiate a redox reaction with SnO2/WO3 NPs. As a result, no gaschromic effects are observed.

Figure S5. Extinction change of PMMA-Ag-SnO2/WO3 fibers exposed to 10,000 ppm n-hexane over exposure time, t min.

Line 305 , page 9 of the revised manuscript 

Based on the observation of CPD changes with and without illumination, we believe that the aforementioned phenomena also occur when the SnO2/WO3 nanoparticles undergo UV/ozone treatment. The possible mechanism is provided as follows. When SnO2/WO3 NPs is conducted the UV/ozone, photo-induced electrons pass through the interface and transfer from WO3 to SnO2, further reacting with O2 to promptly generate superoxide radicals (O2-) (Equation 3). These radicals either adsorb to PMMA or reduce the hexavalent tungsten to pentavalent tungsten (Equation 4-5). The radicals activate the surface of PMMA but weaken the fibers’ structure. When activated PMMA-Ag-SnO2/WO3 fibers are exposed to acetone vapor, the swelling effect diminishes the light scattering of the weaken fiber structure, leading to significant structural deformation and a subsequent decrease in extinction. Furthermore, the incorporation of Ag NPs with surface plasmon resonance further magnifies the decreased extinction behavior, thereby enhancing sensitivity (Figure 3a). On the other hand, under the acetone atmosphere, the collapse PMMA fibers cannot provide plenty of surface adsorbed the O2- radicals, and radicals are also exhausted to acetone degradation (Equation 6-8). These behaviors disrupt the reduction of hexavalent tungsten and cause pentavalent tungsten to release electrons to SnO2 due to the equilibrium heterojunction, converting back to hexavalent tungsten. Consequently, the extinction at wavelengths around 500-600 nm, attributed to WO3, increases, offsetting the overall downward trend. To confirm the mechanism, we further used the chemical inert gas, n-hexane for sensing fibers response. The corresponding extinction spectra and extinction change at varying exposure times are shown in Figure S5. The extinction spectra of sensing fibers remained unaltered upon exposure to n-hexane vapor, suggesting that n-hexane neither deforms the sensing fibers nor influences the light scattering of sensing fiber. Moreover, it does not trigger any gaschromic behavior in heterostructured SnO2/WO3 NPs. Given that PMMA is insoluble in hexane, swelling, and adsorption effects are not anticipated, thereby resulting in no observable changes in light scattering. In addition, the stable alkane structure of n-hexane is not readily influenced by radicals produced from UV-ozone treatment or photo-irradiation of SnO2/WO3. Thus, no redox reactions with SnO2/WO3 nanoparticles occur, and no gaschromic effects are observed. In summary, these observations confirm an enhanced sensitivity to acetone vapor.

Other minor points for consideration:

  1. The term “Core-Shell” Nanocomposite Fiber in the title is a bit misleading as the material is not structured as classical core-shell fibers, nor are the NP´s synthesized real core-shell particles.

Reply:

Thank you for the useful suggestion. We have replaced the term of core-shell WO3@ SnO2 nanocomposite with heterostructure SnO2/WO3 nanocomposite, and updated this modification throughout the manuscript.

  1. In the EDS spectrum presented in Fig. 1i its almost impossible to read or see anything due to small image size and bad resolution. So size and resolution should be increased or the spectrum might be attached in full size as supplementary material

Reply:

Thank you for your comment. We have resized Fig. 1i and enhance the image resolution to ensure that the data is clear and legible, making it more reader-friendly while maintaining the figure's integrity within the manuscript.

Page 5 of the revised manuscript

Figure 1. Characteristic and surface features of pristine WO3 NPs and as-prepared SnO2/WO3 NPs. (a) XRD patterns, (b, c) SEM images, and (d, e) TEM images for WO3 NPs and SnO2/WO3 NPs, (f) high-resolution TEM image of SnO2/WO3 NPs, and the related elemental dispersive spectra in mapping mode, (g) selected area image of nanoparticle, (h) merge image, (i) EDS spectra, (j) oxygen, (k) tungsten, and (l) tin.

  1. Figure 2: Given the width of fiber size distributions, its not meaningful to report the fiber diameters to 0,1 nm, significant digits (also in the text on p. 5).

Reply:

Thank you for your comment. In response to this, we have revised the relevant sections of the manuscript, including the text on page 5 and any corresponding data throughout the document. We have adjusted the reported fiber diameters to reflect an appropriate number of significant digits, removing the numbers after the decimal point. This change aligns the reported data more closely with the practical realities of the measurement variations, ensuring that the information presented is both accurate and scientifically sound.

Line 206, page 5 of the revised manuscript

We then incorporated the as-prepared SnO2/WO3 NPs into the PMMA and Ag NPs precursor solution and proceeded to fabricate a sensing fiber using the electrospinning technique. The photograph of the PMMA-Ag-SnO2/WO3 sensing chip is illustrated in Figure S1a. Pure PMMA-Ag fibers present a regular and uniform fiber morphology with an average diameter of 198 37 nm, and a smooth surface (Figure 2a-b). However, with the incorporation of SnO2/WO3 NPs, the interconnected network structure is reduced, and some beads become embedded into the PMMA-Ag fibers. It leads to the average diameter and relevant deviation of the calculated diameter increase to 204 88 nm (Figure 2c-d). Through EDS mapping observation, we confirmed the presence of embedded particles composed of WO3 and SnO2, as well as the random distribution of SnO2/WO3 NPs in the fibers. (Figure 2e-i). In addition, the UV-ozone treatment was employed to activate the surface of composite fibers before the acetone detection. The outward appearance of the sensing chip after UV-ozone treatment shows insignificant change (Figure S1b).  To confirm the preservation of the three-dimensional network structure of composite fibers after 10 mins of UV-ozone treatment, the morphology observation was undertaken, which is shown in Figure S2. The fibers demonstrated slight shrinkage and their surfaces became rough. Importantly, the integrity of the three-dimensional network structure remained unaltered. This suggests that ozone molecules and oxygen radicals engaged in a mild reaction with the surface of composite fibers.

Figure 2. Characteristics of as-prepared sensing fibers. SEM images of (a) PMMA-Ag fibers and (c) PMMA-Ag-SnO2/WO3 fibers, inset is related low magnification image. The related diameter distribution of (b) PMMA-Ag fibers and (d)PMMA-Ag-SnO2/WO3 fibers. EDS mapping image of PMMA-Ag-SnO2/WO3 fibers, (e) merge image, (f) carbon, (g) tin, (h) tungsten, and (i) oxygen.

  1. Typo in Eq. 4: there should be pentavalent W(V) on the right side of the Eq.

Reply:

We have corrected the equation.

Reviewer 3 Report

Comments and Suggestions for Authors

In this paper (polymers-2668172), the authors fabricated electrospun core-shell WO3@SnO2 nanocomposite fiber for acetone detection. There are certain innovations in material preparation and gas detection, but there are still some unclear results, explanations and technical issues.

1.        Title: In fact, the author tested acetone and did not apply to all VOC gases. Therefore, it is recommended to replace VOC with acetone.

2.        Introduction: (1) “However, the operation requires high temperature and the detection cost is relatively high.” Not strictly stated, in fact, many resistive gas sensors can work at room temperature, even many metal oxide gas sensors can work at room temperature (such as SnO2 for NO2, PEI@SnO2 for acetone). Suggest provide a comprehensive and objective discussion. (2) The research progress of acetone gas sensors needs to be discussed. (3) The research progress of electrospinning in the field of gas sensors should be discussed, such as Mater. Lett. 215 (2018) 58–61.

3.        From the results, the detection limit of 100 ppm seems too high compared to the resistive acetone gas sensor.

4.        What are the issues addressed in this work compared to reference [31]? What are the advantages?

5.        How are acetone gases of different concentrations produced?

6.        How to obtain the response time of 1 min? The authors need to provide the definition of response and response/recovery times of the sensor.

7.        Is the response reversible? Can it be restored?

8.        Suggest providing optical photos of gaschromic properties of sensing materials.

9.        Check the format of the references. The information of some references is incomplete. The mark [] should precede the punctuation mark.

10.    English needs polishing.

Comments on the Quality of English Language

Minor editing of English language required.

Author Response

Reviewer 3

In this paper (polymers-2668172), the authors fabricated electrospun core-shell WO3@SnO2 nanocomposite fiber for acetone detection. There are certain innovations in material preparation and gas detection, but there are still some unclear results, explanations and technical issues.

  1. Title: In fact, the author tested acetone and did not apply to all VOC gases. Therefore, it is recommended to replace VOC with acetone.

Reply:

Thank you for the constructive suggestion. The manuscript’s title has been revised to “Electrospun SnO2/WO3 Heterostructure Nanocomposite Fiber for Enhanced Acetone Vapor Detection”.

  1. Introduction: (1) “However, the operation requires high temperature and the detection cost is relatively high.” Not strictly stated, in fact, many resistive gas sensors can work at room temperature, even many metal oxide gas sensors can work at room temperature (such as SnO2for NO2, PEI@SnO2 for acetone). Suggest provide a comprehensive and objective discussion. (2) The research progress of acetone gas sensors needs to be discussed. (3) The research progress of electrospinning in the field of gas sensors should be discussed, such as Mater. Lett. 215 (2018) 58–61.

Reply:

We have rewritten the Introduction to make the statement more comprehensive.

  1. From the results, the detection limit of 100 ppm seems too high compared to the resistive acetone gas sensor.

Reply:

We agree that our detection limit of 100 ppm is far from the resistive acetone gas sensor. Traditional resistive acetone gas sensors, which are known for their lower detection thresholds. These resistive sensors, often constructed from metal oxides, are highly sensitive due to their significant resistance changes upon gas exposure. However, they also have notable drawbacks, including their susceptibility to humidity and the necessity for complex, expensive manufacturing processes. In contrast, our sensor leverages the simplicity of electrospinning and the hydrophobic nature of PMMA, creating a more affordable and humidity-resistant device. While we concede the higher detection limit, our sensor counters the challenges often associated with resistive sensors, particularly their humidity interference and high production costs.

  1. What are the issues addressed in this work compared to reference [31]? What are the advantages?

Reply:

In reference [31], we speculated the n-n junction of WO3 and SnO2 NPs would improve the sensitivity to VOCs detection. In this study, we have validated the existence of an n-n junction in SnO2/WO3 heterostructure nanocomposite and elucidated its possible mechanism when exposed to acetone vapor.

  1. How are acetone gases of different concentrations produced?

Reply:

Thank you for the comment. We have improved the paragraph of experimental section and added the detailed procedure for the acetone vapor production. To generate acetone vapor at varying concentrations, we introduced a precisely measured volume of acetone into a 100.0 ml glass liner and allowed it to evaporate completely. The volume of acetone for different concentrations were determined using the ideal gas equation of state (PV=nRT). Following evaporation, we sampled the concentrated acetone vapor, and transferred it into a quartz container. At the same time, we calculated the volume needed to dilute the concentrated vapor. Subsequently, upon injecting the acetone vapor, we initiated measurements of the extinction spectra.

Line 152, page 4 of the revised manuscript

2.3. Detection of Acetone Vapor

To examine the sensitivity of PMMA-Ag-SnO2/WO3 sensing fiber to acetone vapor, the sensing chip was placed in the middle of the container for the acquisition of extinction measurement. The specific volume of acetone solution was injected into a 100.0 ml glass liner. The volume for various concentrations of acetone vapor production was calculated by the ideal gas equation of state (PV=nRT). Following evaporation, we sampled the concentrated acetone vapor and transferred it into a 4.5 cm by 4.0 cm by 4.0 cm quartz glass container. At the same time, we calculated the volume needed to dilute the concentrated vapor. Subsequently, upon injecting the acetone vapor, we initiated measurements of the extinction spectra. The extinction spectrum was immediately recorded using a UV-VIS spectrometer, scanning wavelengths from 400 nm to 900 nm, with measurements taken every 30 seconds. The examination of acetone detection was conducted at room temperature (25 ◦C) within a relative humidity range of 60%. In general, the transmittance of the materials results in various extinction intensities. PMMA-Ag-SnO2/WO3 fiber exhibits a broad extinction shoulder at around 410 nm. Hence, we defined the extinction change as the difference in extinction peak observed before and after exposure to VOC vapor. How to obtain the response time of 1 min?

  1. The authors need to provide the definition of response and response/recovery times of the sensor.

Reply:

Thank you for the comment. The response time was defined when the extinction change ( Et) at exposure time (t) was higher than the threshold, which was derived from the maximum extinction change in air.

Line 232 , page 6 of the revised manuscript

To evaluate VOCs detection of PMMA-Ag-SnO2/WO3 sensing fibers, we measured the transmittance using a UV-Vis spectrometer and converted it into extinction spectra. The change in maximum extinction over time corresponds to the detection response. Figure 3a presents the extinction spectra of PMMA-Ag-SnO2/WO3 fibers exposed to 10,000 ppm acetone. The significant downshift of extinction appears at wavelengths ranging from 400 to 500 nm, inferring the fibers' structure deformation and WO3 gaschromic properties as exposed to acetone vapor. The Ag-induced SPR effect magnified the trend of decreased extinction. The outward appearance also exhibits a slight fade when the sensing chip is exposed to acetone (Figure S1c). As for the control sample of PMMA-Ag-WO3 fibers (Figure 3b), the maximum ∆E when exposed to air over time is approximately -0.008. This value serves as the threshold for acetone detection to mitigate the influence of environmental factors.  The response time is identified when the extinction change (∆E) at a given exposure time (t) exceeds the threshold. Furthermore, both PMMA-Ag-WO3 fibers and PMMA-Ag-SnO2/WO3 fibers (Figure 3b,c) exhibit a noticeable ∆E as exposed to acetone vapor. The change in ∆E over time for PMMA-Ag-SnO2/WO3 fibers, when exposed to acetone concentrations of 10,000, 1,000, and 100 ppm, is higher compared to that of PMMA-Ag-WO3 fibers. It indicates the incorporation of SnO2 NPs to fabricate the heterostructured SnO2/WO3 NPs enhances the extinction response and sensitivity of acetone vapor. The reproducibility of the sensing fiber was further validated. The sensing fibers displayed highly consistent changes in extinction, as shown in Figure S3. To further realize the lowest concentration of detection, we collected the several extinction changes of sensing fibers exposed to acetone vapor with various concentrations and depicted the calibration curve (Figure S4). A positive correlation was observed between the vapor concentration and the extinction change (Figure S4a), indicating a strong dependence of the extinction change on the concentration of acetone vapor. Additionally, through the formulation of vapor concentration and the corresponding average extinction change at t = 30 mins, the calibration curve can be defined as follows:

Et= -0.00516 ln(C) -0.016                                                                  (2)

Where ΔE represents the change in extinction at a given exposure time (t), and C denotes the vapor concentration in ppm. Considering the threshold (-0.008) of extinction change in ambient air, the sensing fiber is anticipated to exhibit sensitivity down to a concentration of 0.212 ppm. Consequently, the expected detection limit and response time for acetone detection using PMMA-Ag-SnO2/WO3 fibers are notably below 100 ppm and within 1.0 min.

  1. Is the response reversible? Can it be restored?

Reply:

Thank you for the constructive suggestion. In terms of the recyclability of our sensing materials, it's important to note that they exhibit significant and irreversible deformation when exposed to VOCs. However, the unique performance characteristics of these sensing materials, such as high sensitivity, real-time detection, and cost-effectiveness, remain quite remarkable. Consequently, these sensing materials could be considered as disposable sensing chips suitable for practical applications.        

Figure S3. Reproducibility test of PMMA-Ag-SnO2/WO3 fibers exposed to 100 ppm acetone vapor.

  1. Suggest providing optical photos of gaschromic properties of sensing materials.

Reply:

Thank you for the suggestion. We took the photographs of the PMMA-Ag-SnO2/WO3 sensing chip in initial state, post UV-ozone treatment and exposure to acetone vapor, presented in Figure S1. Figure S1a, b reveals a minimal change in appearance with and without UV-ozone treatment. It is noteworthy that the white color of the sensing chip exhibits a slight fade when the sensing chip exposed to acetone vapor (Figure S1c). This phenomenon could be attributed to deformation and the swelling effect within the sensing fiber, which subsequently reduces light scattering. Although no gaschromic behavior attributable to the SnO2/WO3 heterostructure nanocomposite within the fiber is visually evident—likely due to its low concentration—UV-Vis spectrometric analyses reveal significant shifts in extinction across various wavelength bands. This confirms microscale alterations in optical properties.

Figure S1. Photographs of the PMMA-Ag-SnO2/WO3 sensing fibers: (a) in initial state, (b) post UV-Ozone treatment, and (c) exposure to acetone vapor.

Line 206, page 5 of the revised manuscript

We then incorporated the as-prepared SnO2/WO3 NPs into the PMMA and Ag NPs precursor solution and proceeded to fabricate a sensing fiber using the electrospinning technique. The photograph of the PMMA-Ag-SnO2/WO3 sensing chip is illustrated in Figure S1a. Pure PMMA-Ag fibers present a regular and uniform fiber morphology with an average diameter of 198 37 nm, and a smooth surface (Figure 2a-b). However, with the incorporation of SnO2/WO3 NPs, the interconnected network structure is reduced, and some beads become embedded into the PMMA-Ag fibers. It leads to the average diameter and relevant deviation of the calculated diameter increase to 204 88 nm (Figure 2c-d). Through EDS mapping observation, we confirmed the presence of embedded particles composed of WO3 and SnO2, as well as the random distribution of SnO2/WO3 NPs in the fibers. (Figure 2e-i). In addition, the UV-ozone treatment was employed to activate the surface of composite fibers before the acetone detection. The outward appearance of the sensing chip after UV-ozone treatment shows insignificant change (Figure S1b).  To confirm the preservation of the three-dimensional network structure of composite fibers after 10 mins of UV-ozone treatment, the morphology observation was undertaken, which is shown in Figure S2. The fibers demonstrated slight shrinkage and their surfaces became rough. Importantly, the integrity of the three-dimensional network structure remained unaltered. This suggests that ozone molecules and oxygen radicals engaged in a mild reaction with the surface of composite fibers.

  1. Check the format of the references. The information of some references is incomplete. The mark [] should precede the punctuation mark.

Reply:

We have carefully checked the format of the references.

  1. English needs polishing.

Reply:

The manuscript has been carried out language edition by a native English speaker and corrected the grammatical mistakes in the revised manuscript.

Round 2

Reviewer 1 Report

Comments and Suggestions for Authors

TEM image of the composite nanofibers before and after the UV-ozone should be provided. The quality of the provided SEM image is quite low.

Comments on the Quality of English Language

The quality of English should be further improved.

Author Response

Reviewer 1
TEM image of the composite nanofibers before and after the UV-ozone should be provided. The quality of the provided SEM image is quite low.

Reply:

Thank you for your comment. We have added the TEM images of PMMA-Ag-SnO2/WO3 fibers with and without UV-ozone treatment in supporting information.

Reviewer 2 Report

Comments and Suggestions for Authors

The authors considered the reviewer comments. They added missing data and discussions and have thus significantly improved the manuscript. I would consider it appropriate for publication in polymers.

Comments on the Quality of English Language

The manuscript still contains some flaws in english and some further language polishing would be advisable

Author Response

Reviewer 2

The authors considered the reviewer comments. They added missing data and discussions and have thus significantly improved the manuscript. I would consider it appropriate for publication in polymers.

Reply:

We appreciate the reviewer's insightful comments and suggestions, which have helped improve our manuscript.

Reviewer 3 Report

Comments and Suggestions for Authors

The response and revised manuscript are satisfactory, and it is recommended to accept.

Author Response

Reviewer 3

The response and revised manuscript are satisfactory, and it is recommended to accept.

Reply:

We appreciate the reviewer's insightful comments and suggestions, which have helped improve our manuscript.
